# TiKMiX: Take Data Influence into Dynamic Mixture for Language Model Pre-training

## Abstract

The data mixture in language model pre-training is a cornerstone of its final performance. However, a static mixing strategy is suboptimal, as the model's learning preferences for various data domains shift dynamically throughout training. Crucially, observing these evolving preferences in a computationally efficient manner remains a significant challenge. To tackle this, we propose TiKMiX, a method that dynamically adjusts the data mixture according to the model's evolving preferences. Specifically, we introduce Group Influence for TiKMiX, an efficient metric for evaluating the impact of data domains on the language models, which can formulate the data mixing problem as a search for the optimal influence-maximizing distribution. We solve this via two approaches: TiKMiX-D for direct optimization, and TiKMiX-M, which uses a regression model to predict a superior mixture. We train language models with different parameter scales, on up to 1 trillion tokens. TiKMiX-D exceeds the performance of SOTA mixing strategies like REGMIX while using just 20% of the computational resources. TiKMiX-M leads to an average performance gain of 2% across 9 downstream benchmarks. Our experiments reveal that a model's data preferences evolve with training progress and scale, and we demonstrate that dynamically adjusting the data mixture based on Group Influence, a direct measure of these preferences, significantly improves performance by mitigating the "under digestion" of data seen with static ratios.

## 1 Introduction

The availability of large-scale public datasets has been a key factor in the creation of Large Language Models (LLMs). The pre-training data for LLMs is predominantly sourced from the internet Wettig et al. (2025); Yu et al. (2025a), encompassing a wide range of materials such as academic papers Tirumala et al. (2023), books Tirumala et al. (2023), and more. The mixture ratio of data from different domains plays a crucial role in determining the capabilities of large language models (LLMs) Zhang et al. (2025b); Liu et al. (2025b); Bai et al. (2024a). For example, the developers of GPT-3 Floridi & Chiriatti (2020) regard Wikipedia as a source of exceptionally high-quality data and increase its proportion within the training dataset. REGMIX Liu et al. (2024) leverages results from small-scale experiments to automatically set its mixing ratios; however, it does not take into account dynamic changes in the model's state during training Yu et al. (2024); Zhang et al. (2025a). This observation raises a critical research question: *How can we dynamically select training data for a model in accordance with its preferences, while ensuring both scalability and efficiency?*

Prior research Xie et al. (2023); Fan et al. (2023); Team (2024); Albalak et al. (2023) has leveraged small proxy models to determine domain weights for large-scale language models. However, this approach is computationally expensive, as it requires training proxy models on massive datasets, often exceeding 100 billion tokens. Some methods assume that the relative performance of data mixtures remains stable across different model scales and training durations Liu et al. (2024); nevertheless, they overlook the dynamic nature of a model's data preferences as training progresses. Approaches such as ODM Albalak et al. (2023) attempt to address this issue by monitoring training dynamics to guide data allocation, but their iterative nature is inefficient when dealing with the ever-increasing scale of pre-training data Jin et al. (2024); Wang et al. (2025). A significant gap remains in current practices: leading LLMs Yang et al. (2025); Team et al. (2025); Dubey et al. (2024) typically employ multi-stage pre-training, yet lack mechanisms for rapid and dynamic data re-weighting between stages that can adapt to the model's evolving preferences.

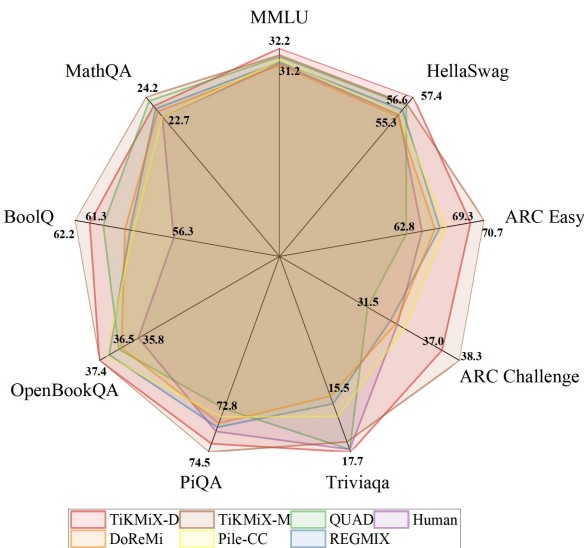

Figure 1: Performance comparisons of our TiKMiX with current state-of-the-art data mixing strategies for pre-training a 1B parameter Language Model with 1T tokens.

We propose a data mixing strategy that dynamically adjusts data proportions during training while incurring minimal computational overhead. Specifically, we introduce **Group Influence**, an efficient method for evaluating the collective impact of each domain on validation performance at low computational cost by leveraging gradient accumulation. This approach enables quantification of the model's data preferences at any stage of training. Building on this foundation, we present **TiKMiX**, a method that formulates dynamic data mixing as an optimization problem: identifying the data combination that maximizes positive influence. To solve this, we develop two variants: **TiKMiX-D**, which directly optimizes a weighted sum of influences from individual domains to determine optimal mixing ratios; and the more advanced **TiKMiX-M**, which uses the output of TiKMiX-D as an initialization, performs perturbation experiments in its vicinity, and fits a regression model to characterize the relationship between mixing ratios and performance, thereby predicting a globally optimal mixture for subsequent large-scale training.

With the proposed TiKMiX framework, we are able to dynamically adjust the data mixture strategy throughout the entire pre-training cycle, adapting to changes in both model scale and training stage. In line with previous work Bai et al. (2024b); Kang et al. (2024); Diao et al. (2025); Tao et al. (2025), we conducted experiments on models with varying parameter sizes and scaled training up to 1 trillion tokens. TiKMiX-D surpasses state-of-the-art methods such as REGMIX, achieving comparable or superior performance while requiring only 20% of the computational resources. TiKMiX-M further yields an average performance improvement of 2% across nine downstream benchmarks, as illustrated in Fig. 1. Additionally, we discuss the feasibility and implications of applying TiKMiX to even larger-scale models. Our experiments reveal several key findings: (1) a model's data preferences evolve as training progresses; (2) models of different scales exhibit distinct patterns of preference change; (3) dynamic adjustment of the data mixture facilitates more comprehensive learning of the data by the model. In summary, the main contributions of this paper are as follows:

- We propose **Group Influence**, a novel and efficient method for observing and quantifying the dynamic preferences of Large Language Models for different data domains during the pre-training process.

- We designed **TiKMiX**, a dynamic data mixture framework that leverages the observations from Group Influence to adaptively adjust data ratios, aiming to balance the model's performance across multiple tasks.

- Extensive experiments demonstrate that our method not only significantly enhances model performance but also provides profound insights into how a model's data preferences evolve with the training process and model scale, thereby validating the effectiveness of dynamically adjusting data proportions.

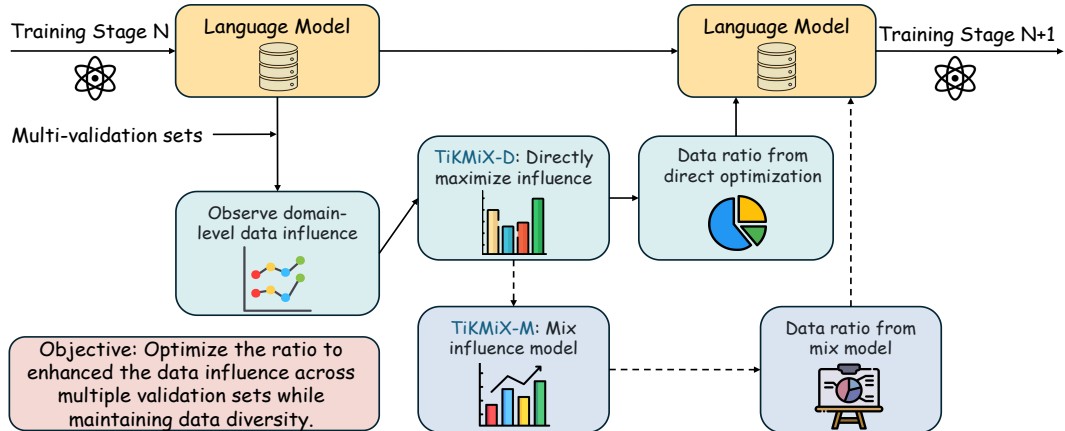

Figure 2: The process involves periodically measuring domain contributions via Group Influence and adjusting the data mixture to maximize learning efficiency.

## 2 RELATED WORK

### 2.1 INFLUENCE FUNCTION

Influence Functions offer a mathematically grounded method to estimate the effect of training data on model predictions without costly retraining Koh & Liang (2017). Their application to high-dimensional models like Large Language Models (LLMs) has been hampered by the computational challenge of inverting the Hessian matrix. Recent work has overcome this barrier through scalable approximation techniques. Notably, the work by Anthropic Grosse et al. (2023) adapted EK-FAC George et al. (2018), an efficient Hessian approximation, to successfully apply influence functions to 50B-parameter Transformer models. This breakthrough established influence functions as a viable tool for performing data attribution at the scale of modern LLMs, enabling the identification of specific pre-training data that drives model outputs Kou et al. (2025); Choe et al. (2024); Lin et al. (2024a). However, computation at the sample level incurs prohibitive overhead in large-scale pre-training scenarios. Therefore, we propose Group Influence, a method that extends influence functions to groups of data. By leveraging gradient accumulation techniques, Group Influence can efficiently evaluate the collective impact of an entire data domain with relatively low computational cost. This allows us to quantify the model's current data preferences.

### 2.2 DATA SELECTION AND MIXING

Strategic curation of training data significantly enhances model performance Koh & Liang (2017); Albalak et al. (2023). For pre-training Large Language Models (LLMs), data curation methods are commonly categorized by granularity: **Token-level Selection:** The most fine-grained approach, which filters individual tokens according to specific criteria Lin et al. (2024b). **Sample-level Selection:** Methods include heuristic-based approaches Sharma et al. (2024); Soldaini et al. (2024) and learning-based techniques employing optimization algorithms Chen et al. (2024); Shao et al. (2024). Additionally, approaches such as MATES Yu et al. (2024) utilize model-derived signals to inform selection Marion et al. (2023); Ankner et al. (2024). **Group-level Selection:** Earlier work relied on manually defined ratios, while recent advances favor learning-based strategies. Offline methods like REGMIX Liu et al. (2024) and DoReMi Xie et al. (2023) use proxy models to assign static group weights, whereas dynamic methods such as Quad Zhang et al. (2025a) and ODM Albalak et al. (2023) iteratively adjust weights during training. Current mainstream pre-training pipelines are typically divided into multiple stages but often lack a mechanism to dynamically adjust the data mixture ratio based on the model's state in different stages. Our proposed method, TiKMiX, is a semi-offline, group-level selection approach that dynamically adjusts the data mixture ratio across multiple training stages. Unlike fully dynamic methods that require repeated iterative updates, TiK-MiX directly optimizes the mixture ratio based on the model's current data preferences, enabling efficient adaptation without multiple rounds of adjustment.

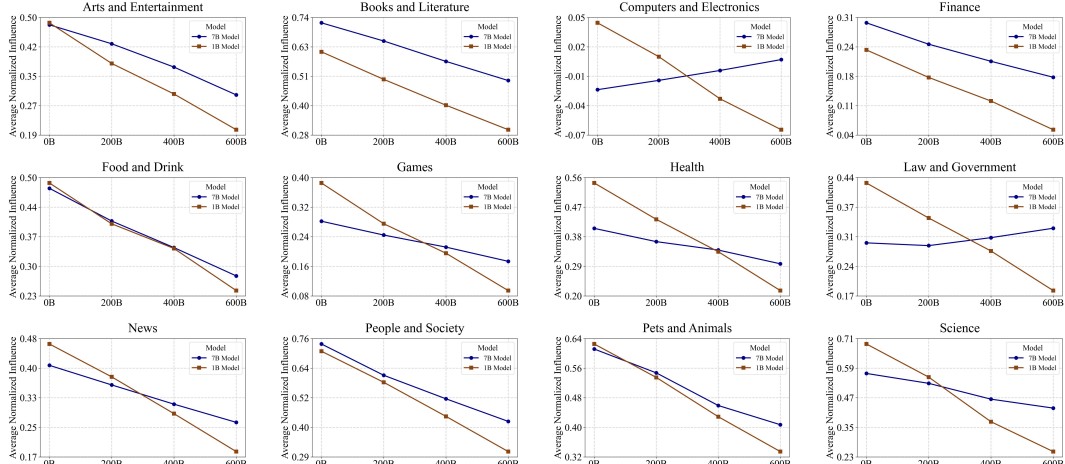

Figure 3: The influence of different domains on the validation set as the model training progresses.

## 3 METHODOLOGY

In this section, we introduce TiKMiX, a framework for dynamically optimizing the data mixture during large language model pre-training as shown in Fig. 2. Our approach is centered on a novel metric, Group Influence, designed to efficiently measure the real-time contribution of each data domain to the model's learning. We formulate the dynamic data mixture problem as an optimization task aimed at maximizing this Group Influence. To solve this, we propose two distinct methods : TiKMiX-D, which directly optimizes the mixture based on influence scores, and TiKMiX-M, which leverages a regression model for a computationally efficient approximation. We first define the problem setup and Group Influence, then elaborate on these two optimization strategies.

### 3.1 GROUP INFLUENCE

Group Influence extends the classical influence function framework from individual data points to cohesive groups of data. We first establish the theoretical motivation for this extension, then provide a mathematical derivation of Group Influence, and finally, discuss its computational properties.

Influence functions offer a principled and computationally efficient method for estimating the effect of a single training instance on a model's parameters or predictions Koh & Liang (2017). By approximating the change in model parameters resulting from upweighting a training point $z$, they provide valuable insights into model behavior without the need for retraining. However, many complex model behaviors, such as systemic bias, factual recall, or vulnerability to specific adversarial attacks, are not attributable to a single, isolated training example. Instead, they often emerge from the collective effect of a *group* of semantically related instances. A linear summation of individual influence scores, i.e., $\sum_{z_i \in S} I(z_i)$, is insufficient as it fails to capture the non-trivial interactions between data points during optimization. The collective gradient of a group can shape the loss landscape in a manner distinct from the sum of its constituent parts. To quantify the consolidated impact of a data subset $S$ as a single entity, we define the Group Influence function. Let a model, parameterized by $\theta \in \mathbb{R}^d$, be trained on a dataset $D = \{z_1, \ldots, z_N\}$ by minimizing an empirical risk objective $J(\theta)$:

$$\theta^* = \arg\min_\theta J(\theta) = \arg\min_\theta \frac{1}{N} \sum_{i=1}^{N} \mathcal{L}(z_i, \theta), \tag{1}$$

where $\mathcal{L}(z_i, \theta)$ is the loss function for instance $z_i$. To measure the influence of a subset $S \subseteq D$, we introduce a perturbed objective where every member of $S$ is simultaneously upweighted by an infinitesimal positive value $\epsilon$. The new optimal parameters $\theta_\epsilon^*$ are found by minimizing this perturbed

objective:

$$\theta_\epsilon^* = \arg\min_\theta \left( \frac{1}{N} \sum_{i=1}^N \mathcal{L}(z_i, \theta) + \epsilon \sum_{z_j \in S} \mathcal{L}(z_j, \theta) \right). \tag{2}$$

This formulation models a scenario where the training process is nudged to place greater emphasis on the group $S$. For $\epsilon = 0$, we recover the original optimal parameters, $\theta_{\epsilon=0}^* = \theta^*$. The influence of group $S$ on the model parameters is then defined as the rate of change of $\theta_\epsilon^*$ with respect to $\epsilon$, evaluated at $\epsilon = 0$. A closed-form expression for this quantity can be derived using the implicit function theorem. The first-order optimality condition for any $\epsilon$ requires that the gradient of the perturbed objective at its minimum $\theta_\epsilon^*$ is zero, which can be formulated as:

$$\nabla_\theta J_\epsilon(\theta_\epsilon^*, S) = \frac{1}{N} \sum_{i=1}^N \nabla_\theta \mathcal{L}(z_i, \theta_\epsilon^*) + \epsilon \sum_{z_j \in S} \nabla_\theta \mathcal{L}(z_j, \theta_\epsilon^*) = 0. \tag{3}$$

Differentiating this entire equation with respect to $\epsilon$ via the chain rule yields:

$$\frac{d}{d\epsilon}\left[\nabla_\theta J_\epsilon(\theta_\epsilon^*, S)\right] = \nabla_\theta^2 J_\epsilon(\theta_\epsilon^*, S)\frac{d\theta_\epsilon^*}{d\epsilon} + \frac{\partial}{\partial\epsilon}\left(\nabla_\theta J_\epsilon(\theta_\epsilon^*, S)\right) = 0. \tag{4}$$

Evaluating this expression at $\epsilon = 0$ (where $\theta_{\epsilon=0}^* = \theta^*$), the Hessian $\nabla_\theta^2 J_\epsilon(\theta_\epsilon^*, S)$ simplifies to the Hessian of the original objective, $H_{\theta^*} \triangleq \nabla_\theta^2 J(\theta^*)$. The partial derivative term becomes $\sum_{z_j \in S} \nabla_\theta \mathcal{L}(z_j, \theta^*)$. Substituting these into Equation 4 gives:

$$H_{\theta^*} \left.\frac{d\theta_\epsilon^*}{d\epsilon}\right|_{\epsilon=0} + \sum_{z_j \in S} \nabla_\theta \mathcal{L}(z_j, \theta^*) = 0. \tag{5}$$

Assuming the Hessian $H_{\theta^*}$ is positive definite and thus invertible, we can solve for the influence of group $S$ on the model parameters:

$$I_{\text{param}}(S) \triangleq \left.\frac{d\theta_\epsilon^*}{d\epsilon}\right|_{\epsilon=0} = -H_{\theta^*}^{-1}\left(\sum_{z_j \in S} \nabla_\theta \mathcal{L}(z_j, \theta^*)\right). \tag{6}$$

A common practical application is to measure the influence of $S$ on a scalar-valued function of the parameters, $f(\theta)$, such as the loss on a test sample, $f(\theta) = \mathcal{L}(z_{\text{test}}, \theta)$. By applying the chain rule, the influence of $S$ on $f$ is given by:

$$I_f(S) \triangleq \left.\frac{df(\theta_\epsilon^*)}{d\epsilon}\right|_{\epsilon=0} = \nabla_\theta f(\theta^*)^T \left.\frac{d\theta_\epsilon^*}{d\epsilon}\right|_{\epsilon=0}. \tag{7}$$

Substituting Equation 6 into Equation 7 yields the final expression for the **Group Influence function**, which can be formulated as:

$$I_f(S) = -\nabla_\theta f(\theta^*)^T H_{\theta^*}^{-1}\left(\sum_{z_j \in S} \nabla_\theta \mathcal{L}(z_j, \theta^*)\right). \tag{8}$$

The scalar value $I_f(S)$ quantifies the extent to which upweighting the group $S$ during training would increase ($I_f(S) > 0$) or decrease ($I_f(S) < 0$) the value of the function $f$. A significant computational advantage of Equation 8 is its structure. The $\sum_{z_j \in S} \nabla_\theta \mathcal{L}(z_j, \theta^*)$ is the accumulated gradient of the group $S$. This allows for an efficient implementation where the gradients for all samples within the group are first computed and aggregated. Subsequently, the computationally intensive Hessian-inverse-vector product is performed only once. This structure ensures the computation of Group Influence is scalable, as its cost is not dominated by the cardinality of the group $|S|$.

## 3.2 TIKMIX-D: DIRECTLY MAXIMIZE INFLUENCE

Based on the Group Influence metric, which quantifies the effect of each data domain on model performance, we aim to optimize the data mixture by determining a weight vector $w$ that maximizes

overall influence. We propose TiKMiX-D, which formulates this as a multi-objective optimization problem, dynamically adjusting $w$ during training to balance performance and maintain data diversity. The Group Influence scores are organized into an $n \times m$ matrix $S$, where $n$ is the number of validation tasks and $m$ is the number of data domains, with $S_{ij}$ denoting the influence of domain $d_j$ on task $v_i$. The expected influence for each task is $P = S \cdot w$, and to ensure comparability across tasks, we normalize as follows:

$$\hat{P}_i = \frac{P_i}{\max_j S_{ij} + \epsilon}, \tag{9}$$

$\epsilon$ denotes a small positive constant added for numerical stability. The optimization objective of TiKMiX-D is defined as a unified function $L(w)$ that integrates three components: (1) **influence uniformity**, measured by the standard deviation $\mathrm{std}(\hat{P})$, promoting balanced improvements across tasks; (2) **overall influence gain**, quantified by the sum $\sum \hat{P}_i$, to maximize aggregate performance; and (3) **data diversity**, measured by the entropy $H(w) = -\sum_{j=1}^{m} w_j \log(w_j)$, encouraging a uniform weight distribution. The trade-offs among these objectives are controlled by hyperparameters $\alpha, \beta$, and $\gamma$, which are set to 1 in our experiments for equal weighting.

The complete optimization problem is subject to several constraints to ensure a valid and beneficial solution. The weights must be non-negative ($w_j \geq 0$) and sum to one ($\sum w_j = 1$). Furthermore, to guarantee continuous improvement, we enforce a Pareto improvement constraint, ensuring that the influence generated by the new mixture $w$ is no less than that of the prior mixture $w_{\mathrm{prior}}$ for any task, i.e., $S \cdot w \geq S \cdot w_{\mathrm{prior}}$. This leads to the final constrained non-linear optimization problem:

$$
\begin{aligned}
\underset{w}{\text{minimize}} \quad & \alpha \cdot \mathrm{std}(\hat{P}) - \beta \cdot \sum_{i=1}^{n} \hat{P}_i - \gamma \cdot H(w) \\
\text{subject to} \quad & \sum_{j=1}^{m} w_j = 1, \quad w_j \geq 0 \ \forall j \in \{1, \dots, m\}, \quad S \cdot w \geq S \cdot w_{\mathrm{prior}}.
\end{aligned}
\tag{10}
$$

We employ the Sequential Least Squares Quadratic Programming algorithm Gupta & Gupta (2018) to solve this problem, initializing the weights with a uniform distribution. The resulting optimal vector, $w_{\mathrm{best}}$, serves as the dynamic data mixture for the subsequent training stage.

## 3.3 TiKMiX-M: Mix influence model

While TiKMiX-D provides an efficient strategy for data mixing through direct optimization, it operates on the assumption that the influences of data domains are linearly additive. This simplification may overlook the mix of different domain, non-linear cross-domain interactions that arise when different data sources are combined. We introduce TiKMiX-M, optimize mixture proportions by modeling the interactions within domain mixtures To more accurately capture these mixture effects.

To explore the model's performance across a diverse range of domain weightings, we generate a set of $N$ candidate mixture vectors. Our approach is anchored by an empirically determined prior weight vector, $w_{\mathrm{orig}} \in \mathbb{R}^D$, where $D$ is the number of domains. For each domain $i$, we define a plausible sampling interval by scaling the original weight. We employ Latin Hypercube Sampling Loh (2021) within this $D$-dimensional hyperrectangle to efficiently generate candidate vectors, ensuring a uniform and non-collapsing coverage of the parameter space.

Each candidate vector $w_{\mathrm{cand}}$ is subsequently normalized to satisfy the constraint ($\sum_{i=1}^{D} w_i = 1$), yielding a normalized vector $w_{\mathrm{norm}} = w_{\mathrm{cand}} / \sum_{j=1}^{D} w_{\mathrm{cand},j}$. However, this normalization can shift components outside their predefined intervals. Therefore, we implement a rejection sampling scheme, where a normalized vector $w_{\mathrm{norm}}$ is accepted into our final set only if it satisfies the boundary constraints for all dimensions, i.e., $w_{\mathrm{norm},i} \in [l_i, h_i]$ for all $i \in \{1, \dots, D\}$. This iterative process is repeated until $N$ valid weight vectors that meet both the summation and boundary conditions have been collected, resulting in a robust and well-distributed set of weights for subsequent analysis. For each generated candidate mixture $w_i$, we calculate its true aggregate influence score, $y_i$, across all validation sets using the Group Influence evaluation method $\sum \hat{P}_i$ .

---

**Algorithm 1** Iterative Search via TiKMiX-M

---

**Input:** Surrogate $f_{\text{sur}}$, initial mix $w^{(0)}$, iters $T$, samples $N$, exploration $[\alpha_{\min}, \alpha_{\max}]$, top-$k$.
**Output:** Optimized mixture $w^*$.
$w_{\text{best}} \leftarrow w^{(0)}$
Generate exploration strengths $\{\alpha_t\}_{t=1}^T$ logarithmically from $\alpha_{\max}$ to $\alpha_{\min}$.
**for** $t = 1$ to $T$ **do**
    Sample $N$ domain mixture candidates $\{w_i\}$ around updated $w_{\text{best}}$ via Dirichlet.
    For each sampled $w_i$, compute its Group influence score $y_i = f_{\text{sur}}(w_i)$.
    Select indices $I_{\text{top-k}}$ of $k$ mixtures with highest Group influence scores.
    Update $w_{\text{best}} = \frac{1}{k} \sum_{i \in I_{\text{top-k}}} w_i$.
**end for**
**return** $w_{\text{best}}$

---

Following these steps, we obtain a training set $D_{\text{train}} = \{(w_i, y_i)\}_{i=1}^N$. Inspired by REGMIXLiu et al. (2024), we select LightGBM Ke et al. (2017), an efficient gradient boosting decision tree model, as our regression surrogate. This model, $f_{\text{LGBM}}$, is trained to predict the aggregate influence $y$ for given data mixture $w$, i.e., $y = f_{\text{LGBM}}(w)$. We leverage it to efficiently explore the mixture space without performing expensive, true influence evaluations. We design an iterative search algorithm that balances exploration and exploitation to find the optimal mixture.

The process is detailed in Algorithm 1. We start from the ratio from TiKMiX-D, $w_{\text{best-D}}$. At each step, we sample candidate mixtures on the current best solution. The distribution's concentration parameter is annealed over steps, beginning with a large value to encourage global exploration and gradually decreasing to promote local exploitation near the optimum. We employ the surrogate model to evaluate all sampled candidates. The center for the next iteration is then updated to be the average of the top-k candidates with the highest predicted scores. This procedure is repeated until convergence or a maximum number of iterations is reached. TiKMiX-M not only accounts for non-linear cross-domain interactions but also significantly enhances search efficiency through the surrogate model, enabling it to discover superior solutions within the vast mixture space.

## 4 EXPERIMENTS

This section presents a comprehensive set of experiments designed to validate the effectiveness of our TiKMiX framework. We first outline the experimental setup, including evaluation benchmarks, datasets, and baseline methods. Subsequently, we demonstrate that: (1) the pre-training data mixture significantly impacts downstream task performance; (2) our proposed Group Influence is an effective predictor of downstream performance; and (3) the TiKMiX framework, particularly TiKMiX-D and TiKMiX-M, markedly improves model performance and surpasses existing SOTA methods.

### 4.1 EXPERIMENTAL SETUP

**Datasets and Models** Optimizing the data mixture of web-scale corpora is a crucial and highly impactful step in pre-training performant LLMs. While the diversity of web data presents unique challenges, effective mixing strategies can unlock significant performance gains. To systematically investigate this, we conduct our experiments on the RefinedWeb dataset Penedo et al. (2023), which comprises 26 distinct data domains. Following the baseline experimental setup, we adopt the model architecture proposed by Zhang et al. (2024) and construct models with 1B and 7B parameters and train on up to 1 trillion tokens. The training process is divided into two distinct stages, each consisting of 500 billion tokens, with a strategic adjustment of the data mixture ratio at the transition point between stages. We compare TiKMiX against several representative data mixing strategies: **Pile-CC Gao et al. (2020)**: The original data mixture proposed by the authors of The Pile based on heuristics. **REGMIX Liu et al. (2024)**: SOTA method that uses a regression model to predict and optimize validation loss for determining the mixture. **DoReMi Xie et al. (2023)**: a classic dynamic data mixing method that relies on a proxy model. **QUAD Zhang et al. (2025a)**: a method for dynamic selection during training after clustering data We use the best-reported mixture from their paper, re-normalized to the domains available in our setup.

**Downstream Task Evaluation**  To comprehensively evaluate our proposed method, we curated a diverse set of 9 widely recognized downstream benchmarks, which were strategically divided into two categories: in-domain and out-of-domain. This division allows for a rigorous assessment of both the model's core capabilities and its generalization prowess. Our **in-domain** evaluation suite was designed to cover a wide spectrum of reasoning and knowledge-based tasks. It includes **MMLU** Hendrycks et al. (2020), a challenging benchmark measuring knowledge; **HellaSwag** Zellers et al. (2019), a commonsense reasoning task that involves choosing the most plausible continuation for a given context; **ARC** Clark et al. (2018), which we evaluate on both the Easy (**ARC-E**) and the more difficult Challenge (**ARC-C**) sets of grade-school science questions; and **TriviaQA** Joshi et al. (2017), a reading comprehension benchmark requiring models to locate answers within lengthy documents. To evaluate the generalization capabilities of our method, we selected a set of out-of-domain benchmarks. These include **PiQA** Bisk et al. (2020), a commonsense benchmark focused on physical interactions; **OpenBookQA** Mihaylov et al. (2018), a question-answering task requiring reasoning over a given set of science facts; **BoolQ** Clark et al. (2019), a dataset of naturally occurring yes/no questions; and **MathQA** Amini et al. (2019), a mathematical reasoning benchmark with multi-step word problems.

## 4.2 GROUP INFLUENCE AS AN EFFECTIVE PREDICTOR OF PERFORMANCE

The core hypothesis of our introduced TiKMiX framework is that maximizing Group Influence can effectively enhance overall downstream task performance. To validate this hypothesis, we calculated the impact of 10 different data mixtures on various benchmarks. As validation, we trained a 1B-parameter model on 500B data using the corresponding mixtures. The normalized scores are shown in Fig. 4. We observe a strong positive correlation (*i.e.*, Pearson correlation coefficient $\rho = 0.789$) between the total Group Influence and the average downstream scores. This indicates that mixtures generating higher total influence almost invariably lead to better downstream performance. This finding not only confirms the validity of Group Influence as an optimization target but also provides a solid theoretical foundation for the design of our proposed TiKMiX-D and TiKMiX-M.

| Benchmark | Human | DoReMi | Average | QUAD | Pile-CC | REGMiX | TiKMiX-D | TiKMiX-M |
|---|---|---|---|---|---|---|---|---|
| *In-Domain Benchmarks* | | | | | | | | |
| MMLU Hendrycks et al. (2020) | 31.3 | 31.2 | 30.9 | 31.7 | 31.2 | 31.5 | **32.2** | 31.8 |
| HellaSwag Zellers et al. (2019) | 55.5 | 55.3 | 55.9 | 56.5 | 55.6 | 56.0 | **57.4** | 56.6 |
| ARC Easy Clark et al. (2018) | 64.4 | 65.7 | 64.1 | 62.8 | 63.2 | 66.2 | 69.3 | **70.7** |
| ARC Challenge Clark et al. (2018) | 33.7 | 33.6 | 32.1 | 33.5 | 32.7 | 33.2 | 37.0 | **38.3** |
| Triviaqa Joshi et al. (2017) | 17.6 | 15.5 | 17.3 | 17.6 | 16.3 | 15.8 | **17.7** | 17.3 |
| *Out-of-Domain Benchmarks* | | | | | | | | |
| PiQA Bisk et al. (2020) | 73.5 | 73.1 | 71.5 | 72.4 | 69.2 | 73.3 | 74.1 | **74.5** |
| OpenBookQA Mihaylov et al. (2018) | 35.8 | 36.5 | 34.6 | 36.6 | 37.1 | 37.0 | 37.4 | **37.4** |
| Boolq Clark et al. (2019) | 56.3 | 59.2 | 58.3 | 60.5 | 58.7 | 58.9 | 61.3 | **62.2** |
| MathQA Amini et al. (2019) | 22.7 | 23.1 | 23.7 | 23.9 | 22.5 | 23.3 | 23.5 | **24.2** |
| Estimated FLOPs | 0 | 4.2e19 | 0 | 2.3e18 | 0 | 3.7e18 | 7.2e17 | 3.2e18 |
| Average Perf. | 43.4 | 43.7 | 43.2 | 43.9 | 42.9 | 43.9 | 45.5 | 45.9 |
| Best On | 0/9 | 0/9 | 0/9 | 0/9 | 0/9 | 0/9 | 4/9 | 6/9 |

Table 1: Comparison of 1B Parameter Models Trained on 1T Tokens Across Various Benchmarks. The best-performing model on each benchmark is highlighted in bold.

Building on the preceding findings, we formally evaluate the two implementations of our TiKMiX framework: TiKMiX-D and TiKMiX-M. We first followed the natural data distribution, then using TiKMiX adjusted the data mixture between two stages during the 1T-token pre-training process. As shown in Table 1, both of our methods significantly outperform all baselines. On average, across 9 benchmarks, TiKMiX-D and TiKMiX-M improved performance by **1.6%** and **2.0%**, respectively, over the strongest baseline, REGMIX. Notably, on challenging tasks like ARC Easy and ARC Challenge, TiKMiX-M achieved a performance advantage of over 4.8%. The results of experiments conducted on larger-scale models are provided in Table 3

## 4.3 ANALYSIS OF COMPUTATIONAL EFFICIENCY

The exact computation of the Hessian matrix in LLMs typically incurs extremely high computational costs. To mitigate this overhead, we draw upon recent studies on influence functions in LLMsGrosse et al. (2023) and employ the Empirical Kronecker-Factored Approximate Curvature

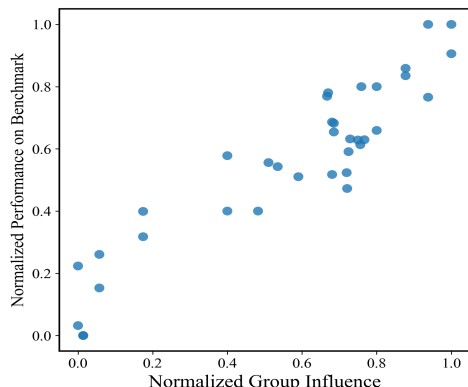

Figure 4: Analysis of the Group Influence and actual performance on the benchmark.

| Benchmark | Loss | | TiKMiX-D | |
|---|---|---|---|---|
| | 5B | 10B | 0.1B | 0.5B |
| *In-Domain Benchmarks* | | | | |
| MMLU Hendrycks et al. (2020) | 31.4 | 31.2 | 32.2 | 32.1 |
| HellaSwag Zellers et al. (2019) | 56.3 | 56.4 | 57.4 | 57.6 |
| ARC Easy Clark et al. (2018) | 67.3 | 65.6 | 69.3 | 69.1 |
| ARC Challenge Clark et al. (2018) | 34.4 | 33.4 | 37.0 | 37.1 |
| TriviaQA Joshi et al. (2017) | 16.5 | 16.9 | 17.7 | 17.9 |
| *Out-of-Domain Benchmarks* | | | | |
| PiQA Bisk et al. (2020) | 73.2 | 73.5 | 74.1 | 74.2 |
| OpenBookQA Mihaylov et al. (2018) | 36.4 | 36.6 | 37.4 | 37.3 |
| BoolQ Clark et al. (2019) | 59.4 | 59.7 | 61.3 | 61.5 |
| MathQA Amini et al. (2019) | 23.9 | 23.7 | 23.5 | 23.6 |
| Average Perf. | 44.3 | 44.1 | 45.5 | 45.6 |

Table 2: The ablation study of Loss and TiKMiX on different data sizes.

(EKFAC) method to approximate the Hessian matrix. EKFAC reduces computational and memory requirements by partitioning the Hessian and applying Kronecker factorization, thereby transforming complex high-dimensional matrix operations into computations within lower-dimensional subspaces.

Consequently, TiKMiX demonstrates superior computational efficiency. In contrast to methods such as MATESYu et al. (2024), Group-MATESYu et al. (2025b), and REGMIX, which require the additional overhead of training small proxy models, the Group Influence calculation and optimization process in TiKMiX is highly efficient and does not involve such auxiliary training procedures. In our 1B model experiments, the total computational overhead for **TiKMiX-D** to determine the next-stage mixture (including influence calculation and regression model inference) was only about **20%** of that required by the **RegMix** method, while achieving comparable or even superior performance. This high efficiency makes TiKMiX a practical and powerful tool for large scale LLM training.

## 4.4 ABLATION STUDY

We conduct a series of ablation studies, with the results presented in Table 2. Our primary investigation focused on the efficacy of using **group influence** and **TiKMiX** for preference observation and data mixture adjustments. As shown in Table 2, our approach allows for the accurate observation of model preferences using only 0.1B tokens and requires no model training, leading to a significant performance improvement over the loss. This highlights the superiority of our method in efficiently identifying and correcting data biases. We further discuss the effectiveness of our model on a larger scale in the appendix.

## 5 CONCLUSION AND DISCUSSIONS

In this work, we address the suboptimality of static data mixing strategies in language model pre-training, demonstrating that a model's learning preferences for different data domains evolve dynamically with its training progress. To tackle this, we introduce TiKMiX, a novel framework that dynamically adjusts the data mixture based on Group Influence, a highly efficient metric to evaluate the contribution of data domains to the model's performance. By framing data mixing as an influence-maximization problem, we developed two approaches: TiKMiX-D, which directly optimizes the mixture and surpasses state-of-the-art methods like REGMIX using only 20% of the computational resources, and TiKMiX-M, which uses a regression model to predict superior mixtures, achieving an average performance gain of 2% across 9 downstream benchmarks. Our experiments confirm that dynamically adjusting the data mixture based on Group Influence significantly improves performance by mitigating the under-digestion of data seen with static ratios. We plan to conduct further experiments on larger-scale models and more diverse datasets to further validate the effectiveness of Group Influence and TiKMiX.

## REPRODUCIBILITY STATEMENT

We provide comprehensive details of the TiKMiX methodology, experimental setup, data processing procedures, and model training specifics in the main text, appendix, and supplementary materials. Specifically, the Methodology section (lines 211–258) systematically presents the mathematical definition and derivation of Group Influence; the appendix further elaborates on the assumptions and provides complete theoretical proofs. The experimental section (lines 365–371) enumerates the datasets used as well as the model architectures and scales adopted in our study. The training process is detailed in lines 419–423, while the Downstream Task Evaluation section (lines 378–392) describes the downstream evaluation benchmarks and baseline comparisons, with evaluation criteria clearly stated in both the main text and appendix. All procedures related to data processing, mixture ratio adjustment, and hyperparameter settings are thoroughly documented in the main text and supplementary materials. The necessary source code is provided via an anonymous downloadable link in the supplementary materials. We believe these resources offer robust support for the research community to reproduce, validate, and further extend our work.

## ETHICS STATEMENT

This study strictly follows the ICLR Code of Ethics, upholds a responsible research attitude, and is dedicated to advancing trustworthy machine learning and artificial intelligence technologies while focusing on their positive impact on society and human well-being. Throughout our work, we fully considered ethical principles such as promoting social welfare, fairness and inclusiveness, scientific integrity, risk prevention, transparency, intellectual property, and privacy protection. All experiments are based on public datasets, with processes that are transparent and reproducible, and there is no fabrication or manipulation of data or results. We strictly adhere to data usage agreements without involving personal privacy.

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

# 6 APPENDIX

## 6.1 EXPERIMENTAL SETUP

**Datasets and Models**  Web data serves as one of the core sources for pre-training large language models (LLMs), playing a crucial role in enhancing model capabilities due to its broad coverage and diversity. However, precisely because web data encompasses a wide range of domains—including news, encyclopedias, forums, and academic content—its highly diverse origins make it extremely challenging to achieve a balanced mixture across different domains. We follow the same experimental setup as prior studies on web data mixture Wettig et al. (2025); Liu et al. (2025a), utilize the RefinedWeb dataset Penedo et al. (2023), and employ the domain classifier He et al. (2023) to categorize the data into 26 distinct domains. Our models, ranging in size from 1B to 7B parameters, are trained on up to 1 trillion tokens. The training process is divided into two distinct stages, each consisting of 500 billion tokens, with a strategic adjustment of the data mixture ratio at the transition point between stages. We compare TiKMiX against several representative data mixing strategies: **Pile-CC Gao et al. (2020)**: The original data mixture proposed by the authors of The Pile based on heuristics. **REGMIX Liu et al. (2024)**: SOTA method that uses a regression model to predict and optimize validation loss for determining the mixture. **DoReMi Xie et al. (2023)**: a classic dynamic data mixing method that relies on a proxy model. **QUAD Zhang et al. (2025a)**: a method for dynamic selection during training after clustering data We use the best-reported mixture from their paper, re-normalized to the domains available in our setup.

Our proposed TiKMiX method achieves a balance between dynamic adaptability and computational efficiency in data mixture strategies. Similar to other dynamic approaches such as DoReMi and QUAD, TiKMiX adjusts the data mixture ratios according to the current state of the model. However, unlike these methods, TiKMiX does not require multiple iterations, which significantly improves training efficiency. Furthermore, TiKMiX simplifies the data mixing process and reduces engineering complexity without sacrificing model performance.

To systematically evaluate the effectiveness of different data mixing strategies, we conduct large-scale experiments on the RefinedWeb dataset. Our models range in size from 1B to 7B parameters and are trained on up to 1 trillion tokens. The training process is divided into two distinct stages, each consisting of 500 billion tokens. At the transition point between these two stages, we strategically adjust the data mixture ratios to further assess the impact of mixing strategies on model performance.

## 6.2 DOWNSTREAM TASK EVALUATION

To conduct a comprehensive and rigorous evaluation of our proposed method, we curated a diverse suite of nine widely-recognized downstream benchmarks. This evaluation matrix is strategically divided into two categories: **in-domain** and **out-of-domain**. This bifurcation allows for a dual-faceted assessment of our model's capabilities: on one hand, to measure its proficiency on tasks closely aligned with its training objectives, and on the other, to critically examine its ability to generalize learned skills to novel tasks and knowledge domains. The consistent performance gains observed across both categories underscore our method's ability to enhance the model's foundational capabilities and foster robust generalization.

**In-Domain Evaluation**  Our in-domain evaluation suite is designed to probe the model's core competencies in complex reasoning, commonsense understanding, and knowledge-intensive applications. These benchmarks are thematically aligned with our method's primary optimization goals and serve to quantify the depth of improvement in these critical areas.

- **MMLU (Massive Multitask Language Understanding)** Hendrycks et al. (2020): A highly challenging multitask benchmark that assesses knowledge across 57 disparate subjects, ranging from elementary mathematics and U.S. history to computer science and law. MMLU demands not only a vast repository of knowledge but also the ability to perform precise, domain-specific reasoning, making it a key indicator of a model's comprehensive intellectual and academic capabilities.
- **HellaSwag** Zellers et al. (2019): A commonsense reasoning benchmark that tasks the model with selecting the most plausible continuation for a given context. HellaSwag is

distinguished by its use of adversarially-generated distractors, which are designed to be highly confusable for models that rely on superficial statistical cues. It therefore serves as a robust test of a model's deeper understanding of causality and everyday situations.

- **ARC (AI2 Reasoning Challenge)** Clark et al. (2018): This benchmark evaluates reasoning and comprehension on grade-school science questions. We assess performance on both its subsets: **ARC-Easy (ARC-E)**, which contains questions often solvable via information retrieval, and the more difficult **ARC-Challenge (ARC-C)**, which requires multi-step reasoning and synthesis of knowledge. Evaluating on both allows for a fine-grained analysis of the model's capabilities, from basic knowledge retrieval to complex scientific inference.

- **TriviaQA** Joshi et al. (2017): A large-scale reading comprehension benchmark where questions are authored by trivia enthusiasts, leading to a high degree of diversity and complexity. The task requires models to locate answers within lengthy, evidence-rich documents, often amidst significant distractor information. It primarily evaluates the model's proficiency in long-context processing, precise information retrieval, and fact verification.

**Out-of-Domain Evaluation**    To rigorously assess the generalization power of our method, we selected a set of out-of-domain benchmarks that are distinct from the in-domain tasks in terms of subject matter, format, or required reasoning skills. Performance on these benchmarks directly reflects the model's ability to transfer its learned meta-skills to new and unseen challenges.

- **PiQA (Physical Interaction QA)** Bisk et al. (2020): A commonsense benchmark focused on physical reasoning. Presented in a question-answering format, it requires the model to understand the properties and affordances of everyday objects (e.g., "How can you cool a cup of water faster?"). PiQA probes the model's intuitive grasp of how the physical world operates, a domain of commonsense distinct from academic knowledge, making it an excellent test of generalization.

- **OpenBookQA** Mihaylov et al. (2018): This benchmark simulates an "open-book" exam, requiring the model to answer questions using a given set of elementary science facts. Success demands not only reading comprehension but, more importantly, the ability to reason over and combine these facts to answer questions whose solutions are not explicitly stated. It critically evaluates the model's capacity for multi-step reasoning and knowledge application within a constrained context.

- **BoolQ (Boolean Questions)** Clark et al. (2019): A dataset of naturally occurring yes/no questions, sourced from real user search queries. The challenge lies in the fact that the relationship between the question and the provided evidence passage is often implicit, requiring sophisticated syntactic and semantic analysis to arrive at a correct Boolean judgment. BoolQ effectively measures the model's fine-grained comprehension of natural, conversational language.

- **MathQA** Amini et al. (2019): A mathematical reasoning benchmark featuring multi-step word problems. The task requires models to parse natural language descriptions, formulate a correct sequence of operations, and execute them to find a solution. Covering a diverse range of mathematical reasoning categories, MathQA is a crucial benchmark for evaluating a model's symbolic reasoning and logical chain-of-thought capabilities, representing a significant test of higher-order cognitive skills.

By systematically evaluating our method across this dual-category, nine-benchmark matrix, we demonstrate that our approach not only enhances performance in core competency areas (as shown by MMLU and ARC-C) but also significantly improves the transfer of these abilities to novel contexts (as evidenced by PiQA and MathQA). This comprehensive improvement across both in-domain and out-of-domain tasks provides strong evidence for the effectiveness and generalizability of our method.

To further investigate the impact of model scale on data utilization, we present a supplementary analysis in Figures 5 to 11. Our key finding is that models of different scales (1B and 7B) exhibit significantly different learning responses and form distinct preferences, even when trained on the exact same data. This phenomenon reveals a complex interplay between data utility and model scale. It provides a solid theoretical foundation for understanding and optimizing the data mixture for models of varying sizes.

Table 3: Ablation study of REGMIX and TiKMiX on 1B and 7B models.

| Benchmark | 1B Model | | 7B Model | |
|---|---|---|---|---|
| | REGMIX | TiKMiX-D | REGMIX | TiKMiX-D |
| *In-Domain Benchmarks* | | | | |
| MMLU Hendrycks et al. (2020) | 31.5 | 32.2 | 40.7 | 41.5 |
| HellaSwag Zellers et al. (2019) | 56.0 | 57.4 | 76.6 | 76.4 |
| ARC Easy Clark et al. (2018) | 66.2 | 69.3 | 78.5 | 78.4 |
| ARC Challenge Clark et al. (2018) | 32.2 | 37.0 | 49.4 | 50.2 |
| TriviaQA Joshi et al. (2017) | 15.8 | 17.7 | 46.4 | 45.3 |
| *Out-of-Domain Benchmarks* | | | | |
| PiQA Bisk et al. (2020) | 73.3 | 74.1 | 79.1 | 79.2 |
| OpenBookQA Mihaylov et al. (2018) | 37.0 | 37.4 | 43.2 | 45.4 |
| MathQA Amini et al. (2019) | 23.2 | 23.5 | 28.8 | 29.9 |
| Average Perf. | 43.9 | 45.5 | 55.3 | 56.0 |

## 6.3 EXPERIMENTS ON MODELS OF DIFFERENT SIZES

Considering computational overhead, for the 7B model, we adopted an experimental design similar to REGMIXLiu et al. (2024), training with 500B tokens in the first stage and 200B tokens in the second stage. Table 3 presents the experimental results of our method on models of different scales. It can be observed that our proposed method significantly outperforms the current state-of-the-art approach, REGMIX, on both in-domain and out-of-domain benchmarks. The performance on the 7B model effectively demonstrates the scalability of our approach. Furthermore, we note that unlike the 1B model, the 7B model's performance on the benchmarks consistently improves throughout the training process. This suggests that the advantage of TiKMiX could be even more pronounced with additional training data.

## 6.4 OBSERVATION OF DATA MIXING WITH GROUP INFLUENCE

To conduct a rigorous analysis of inter-domain interactions during mixed training, we designed an experiment to test the principle of influence additivity. Our hypothesis was that the influence of a mixed dataset on a validation set could be accurately predicted by a weighted sum of the influences from its individual constituent domains. To verify this, we first established a baseline mixing recipe using our TiKMiX-D method. We then systematically explored the local space around this recipe by generating 256 perturbed configurations, created by applying a random scaling factor between 0.5 and 2.0 to each domain's original proportion. After filtering out two sampling outliers, we proceeded with 254 unique data mixture configurations. For each of these 254 points, we sampled a corresponding 0.1B token dataset and measured its direct influence. We then compared this empirical influence value against a predicted influence, which was calculated by summing the pre-computed influences of each individual domain, weighted by their respective proportions in the mixture. As depicted in Fig 13 , this comparison revealed a strong linear correlation. Specifically, the Pearson correlation coefficients on the ARCClark et al. (2018), HellaswagZellers et al. (2019), and TriviaQAJoshi et al. (2017) benchmarks reached 0.845, 0.848, and 0.931, respectively, all of which are statistically highly significant ($p < 0.0001$). This result provides compelling evidence that the outcome of data mixing is highly predictable and can be modeled as a linear combination of inter-domain influences. Consequently, this finding offers a solid empirical justification for the theoretical soundness of our proposed two-stage optimization framework, encompassing both TiKMiX-D and TiKMiX-M.

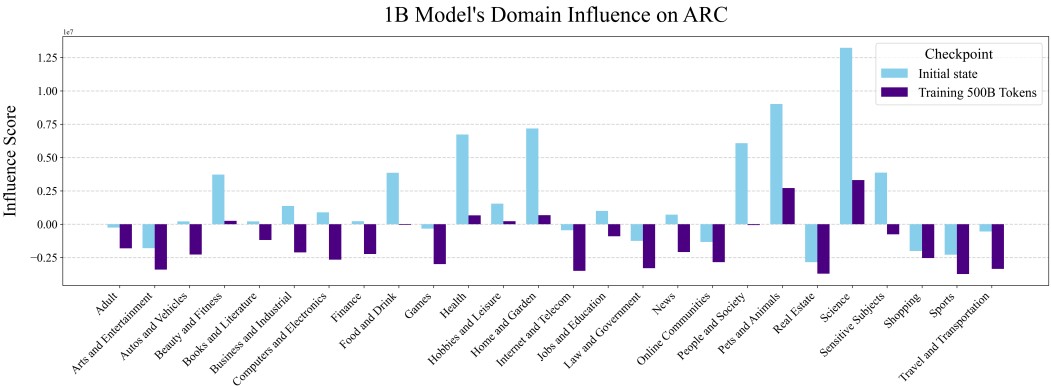

Figure 5: The impact of domains on a 1B model's performance on the ARC benchmark as training progresses.

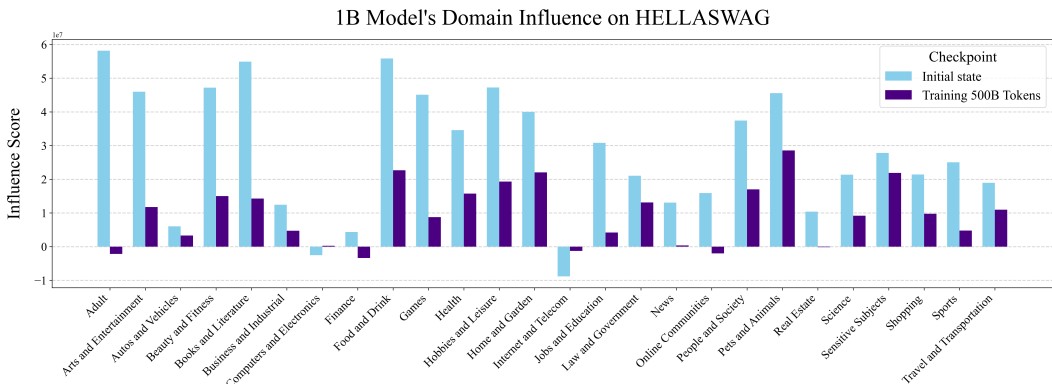

Figure 6: The impact of domains on a 1B model's performance on the HELLASWAG benchmark as training progresses.

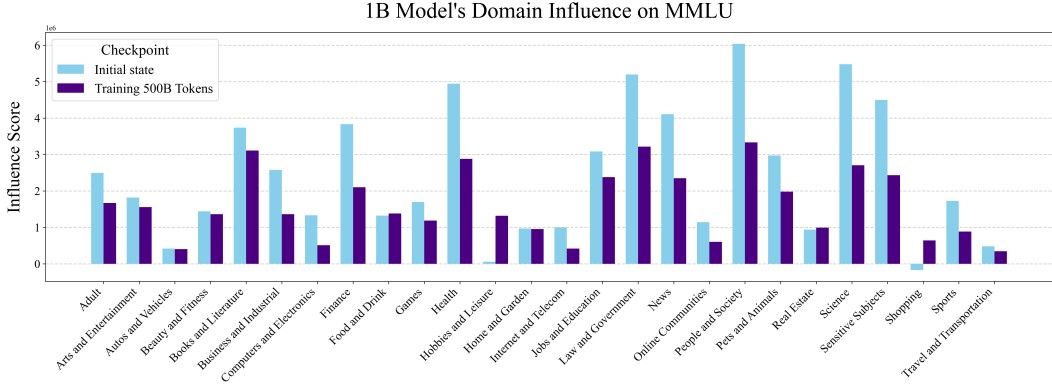

Figure 7: The impact of domains on a 1B model's performance on the MMLU benchmark as training progresses.

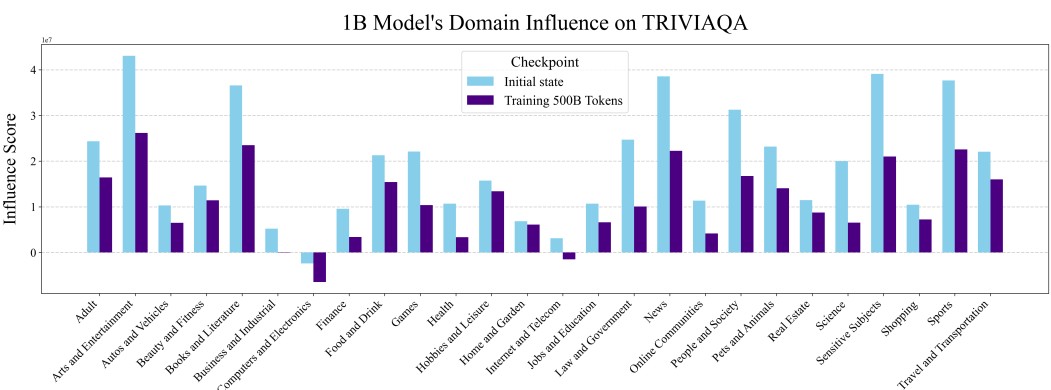

Figure 8: The impact of domains on a 1B model's performance on the TRIVIAQA benchmark as training progresses.

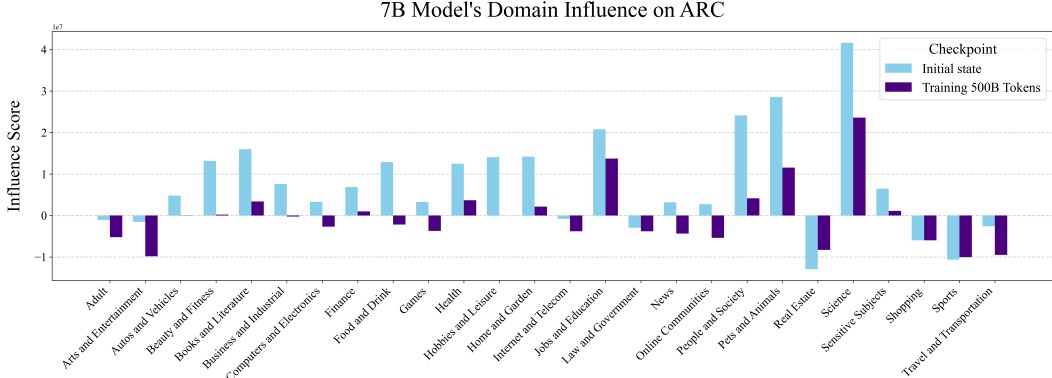

Figure 9: The impact of domains on a 7B model's performance on the ARC benchmark as training progresses.

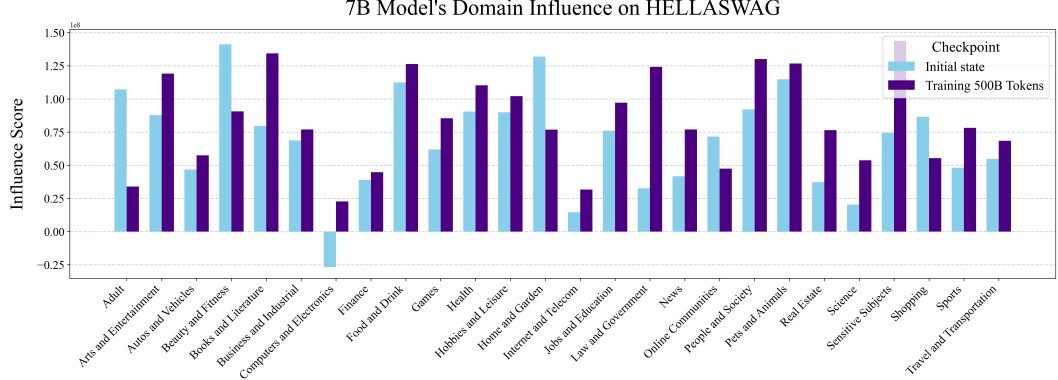

Figure 10: The impact of domains on a 7B model's performance on the HELLASWAG benchmark as training progresses.

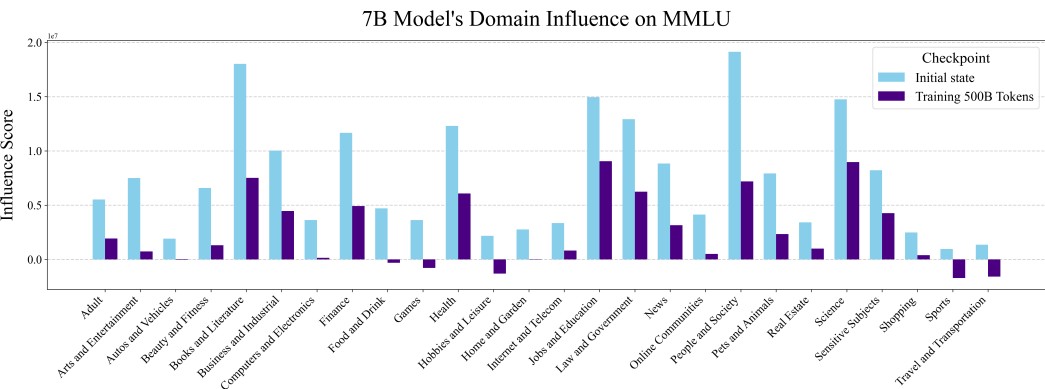

Figure 11: The impact of domains on a 7B model's performance on the MMLU benchmark as training progresses.

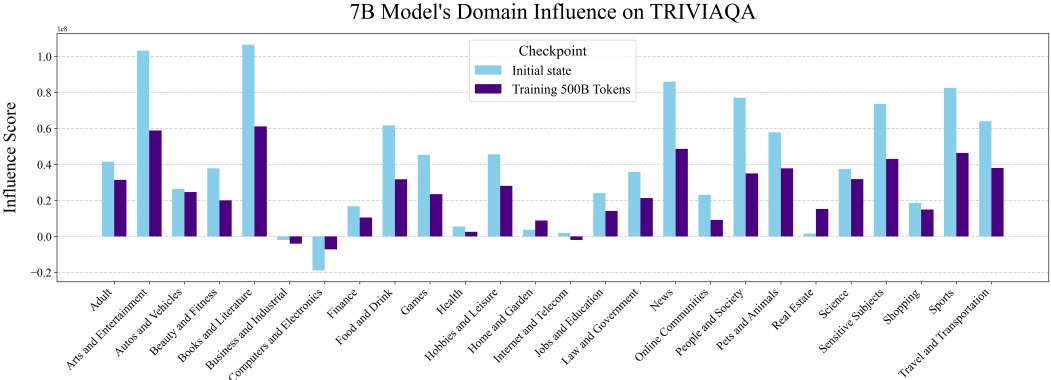

Figure 12: The impact of domains on a 7B model's performance on the TRIVIAQA benchmark as training progresses.

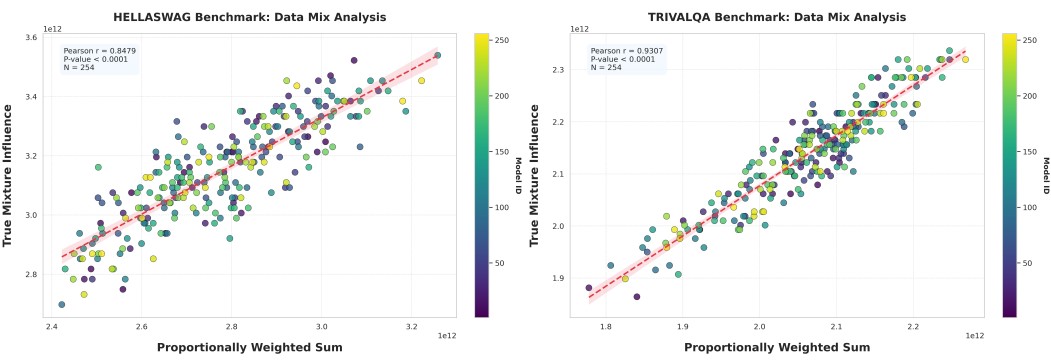

Figure 13: A Group Influence-based Analysis of Data Mixing Effects on Various Benchmarks.

