# OpenReview forum: "TiKMiX: Take Data Influence into Dynamic Mixture for Language Model Pre-training"
_ICLR.cc/2026/Conference — ICLR 2026 Conference Withdrawn Submission_

### Official Review · Reviewer_fbzW · 2025-10-30

**Soundness:** 2
**Presentation:** 3
**Contribution:** 2
**Rating:** 4
**Confidence:** 4

**Summary:**

This paper proposed a new algorithm for data mixture optimization, during the LLM pretraining stage. Compared with previous static data mixing strategies, TikMix algorithm is able to adaptively change the data mixing rates according to the current model status, and improves both the data efficiency and model performance across 9 various benchmarks.

**Strengths:**

1. The paper is well-written and easy to follow;

2. The proposed method demonstrates an improvements on data efficiency and downstream performance across various benchmarks, compared with multiple static data mixing methods.

**Weaknesses:**

1. **The proposed Group Influence is not novel.** [1] already given a comprehensive analysis on the group effect of data influence back in 2019.

2. **Lack of comparison to existing dynamic data mixing strategies**, e.g. DGA[2], which applies a similar strategy but using a first-order gradient alignment as the data influence assessment. Can you provide comparison on both performance and efficiency between the existing dynamic data mixing methods and your proposed ones?

3. **Lack of efficiency analysis.** The proposed method seems to be very costly, which involves KxNxM times of second-order gradient computation at each step to change data mixture rates $w$. (given K as the number of training data domains; M as number of validation tasks; N as the number of mixtures sample at each step). Can you provide detailed cost analysis on the overhead, both from computation and memory aspects?

4. **Concern on scalability**: given the large costs of the data mixing method, it could be hard to be applied on large-scale models. Can you provide the scaling performance analysis on the proposed method?

[1] On the Accuracy of Influence Functions for Measuring Group Effects. Koh et al., 2019.

[2] Dynamic Gradient Alignment for Online Data Mixing. Fan et al, 2024.

**Questions:**

Attached to each point of weakness.

---

### Official Review · Reviewer_DJLX · 2025-10-31

**Soundness:** 2
**Presentation:** 3
**Contribution:** 2
**Rating:** 2
**Confidence:** 4

**Summary:**

TiKMiX introduces a data mixing framework for language model pre-training that adapts data ratios based on the influence of domains on the validation loss. Two methods are proposed: either a model based on influence functions or a boosted tree that regresses the results of experiments with different mixtures with a boosted tree.

**Strengths:**

1. Motivations and related literature are clearly presented.
2. Clean measure of end performance with a clear split into training tasks and the test tasks, not all papers do this.

**Weaknesses:**

1. In Equation (8) can be written as the sum of the influence over the element of the group. It contradicts your motivation L.204 “a linear sum of influence score is insufficient” to introduce group influence in the first place.
2. Why do you need to fit a boosted tree? Once individual group influences are computed a single time, you can then evaluate the influence of any mixture without additional gradient computations, no?

**Questions:**

1. For tikmix-D. At which model \theta do you evaluate group influence?
2. L293. There is little motivation for uniformity and diversity. Why not optimize only overall influence? Why put an equal weight over all three objectives? Do you have empirical evidence that this is a good configuration?
3. I do not understand the PileCC baseline. My understanding is that all models are trained over the refinedweb dataset. How do you apply the Pile CC mixture over the refinedWeb dataset?
4. L368 What are the 26 domains of refinedweb? The original paper/dataset does not mention them, Penedo et al. (2023).
5. I do not understand Table 2. What is ‘Loss’?
6. typos/details
* L049: The first mention of ODM does not define the algorithm.
7. Importance sampling baseline. Have you tried an importance sampling e.g. https://openreview.net/forum?id=p6ncr0eTKE to pick a data distribution close to the test domains?

---

### Official Review · Reviewer_ZUsw · 2025-11-01

**Soundness:** 3
**Presentation:** 2
**Contribution:** 2
**Rating:** 2
**Confidence:** 4

**Summary:**

This paper proposes TiKMiX, a framework for dynamically adjusting data mixture ratios during language model pre-training based on the model's evolving preferences.

This paper introduce
(1) Group Influence: an efficient metric that extends influence functions from individual samples to entire data domains by leveraging gradient accumulation and EKFAC-based Hessian approximation (Equation 8, page 5).
(2) Two optimization approaches: TiKMiX-D directly maximizes a weighted combination of influence uniformity, overall gain, and data diversity, while TiKMiX-M uses LightGBM regression on sampled mixtures to predict globally optimal ratios.

Experiments on 1B-7B parameter models trained on up to 1T tokens from RefinedWeb show TiKMiX-D achieves comparable performance to REGMIX using only 20% computational cost, while TiKMiX-M yields 2% average improvement across 9 benchmarks.

**Strengths:**

1. Strong motivation and practical relevance: The observation that model data preferences evolve during training addresses a real limitation of static mixing strategies used in production LLMs, making this work highly relevant to practitioners.

2. Computational efficiency: Achieving performance gains with only 20% of REGMIX's computational cost by avoiding proxy model training is a significant practical advantage.

3. Comprehensive evaluation: Testing on both in-domain and out-of-domain benchmarks with multiple model scales provides reasonable evidence of generalizability.

**Weaknesses:**

1. Poor writing quality:
    * Mathematical notation is inconsistent (e.g., θ* used for both original and perturbed optima in Equations 1-2)
    * Critical Figure 3 showing evolving domain influences is never referenced or explained in the main text.
    * Dense formula presentation (Section 3.1-3.2) lacks intuitive explanations of why these particular formulations are principled.
    * Minor errors: "To more" (line 308), symbol confusion between Equations 2 and 9. L426: no period symbol.

2. Unclear methodology:
    * Weight vector w ambiguity: The relationship between w in Equations 9/10 and its connection to Equations 8 is never clearly established. Does w represent domain sampling probabilities? How does it relate to the ε-upweighting in Equation 2?
    * Missing architectural details: How is Group Influence computed in practice during training?

3. Limited theoretical insight:
    * The paper claims to be "principled" but doesn't explain why maximizing Group Influence should lead to better generalization beyond empirical correlation.
    * The "free lunch" claim (20% cost, better performance) lacks mechanistic explanation—what fundamental insight makes this possible?

4. Experimental issues:
    * Ablation studies: Table 2 only ablates data size (0.1B vs 0.5B), not the three terms in Equation 10 or the Pareto constraint.
    * Unfair QUAD comparison: QUAD (Zhang et al. 2025a) is a dynamic method that continuously adjusts mixtures during training. You only use its "best-reported mixture" as a static baseline, removing its core dynamic capability.

**Questions:**

1. Figure 3 interpretation: What specific patterns in domain influence evolution (Figure 3) does TiKMiX leverage? Can you identify domains that consistently increase/decrease in importance and explain why this happens? And it seems never be used in the main text?

2. Weight vector formulation: How exactly does the weight vector w in Equation 10 relate to the ε-upweighting in Equation 2? Is w simply the domain sampling probability for the next training stage? Please add explicit notation connecting these.

3. "Free lunch" mechanism: Your method achieves better performance with 20% of REGMIX's cost. What is the fundamental insight that enables this? Is it because:
    * REGMIX's proxy model is unnecessary?
    * Group-level influence is more informative than loss prediction?
    * Something else?

4. Objective function design: Why is the specific formulation in Equation 10 optimal? Have you ablated:
    * Removing the diversity term (γ=0)?
    * Different weightings (α≠β≠γ)?
    * The Pareto improvement constraint?

5. Notation clarification: Equation 2 uses state θ* with ε as perturbed optima, but Equation 9  uses Phat_i with a different ε. Are these the same ε? Please use distinct symbols to avoid confusion.

---

### Official Review · Reviewer_Sdrz · 2025-11-03

**Soundness:** 2
**Presentation:** 3
**Contribution:** 2
**Rating:** 4
**Confidence:** 5

**Summary:**

LLM performance greatly depends on the pre-training data mixture. While most previous methods have focused on learning a static mixture, they typically involve training many small proxy models and cannot capture if the utility of various domains changes as the model is trained. This paper proposes a dynamic data mixing method, TikMix, where group influence scores for each domain are computed across stages of pre-training, and the proposed mix optimizes an average of these group influence scores. A two stage approach that uses TikMix outperforms existing data mixing methods on 26 domains of RefinedWeb. Empirical analysis also shows that the influence scores for each domain changes as model training progresses.

**Strengths:**

Quality:
- The proposed TikMix method outperforms existing mixes and requires less compute.
- Empirical analysis shows that influence scores change throughout training, which motivates the use of dynamic mixing.
- Method is theoreticall motivated, and the optimization problem in equation 10 is well-justified and incorporates various desirable aspects of data (e.g., optimizing the mix while preserving data diversity)

Significance:
- While mixing algorithms that use iterative updates have been previously proposed, the prevailing way of implementing dynamic mixes is through multi-stage training (SmolLM, OLMo2, etc.). However, many technical reports do not explain how to best adjust the mix across these stages or still use heuristics. Therefore, the paper makes great progress towards establishing principles for multi-stage mixing.

**Weaknesses:**

Originality:
- It would be interesting to compare more to existing online approaches; you mention ODM, but here are a few others:
     - Aioli (https://arxiv.org/abs/2411.05735)
     - ADO (https://arxiv.org/abs/2410.11820)
     - These approaches use iterative updates, but that is more of an implementation choice (e.g., using a mirror descent / multiplicative weights style solver versus an exact solver). It would be interesting to see how Aioli---namely, doing some online "re-exploration" of the mixture weight space at 500B tokens and then re-fitting a linear model---would do, since that paper only looks at very small models / token budgets.
- Also, in terms of the linearity of group influence, a related paper is Data Mixing Laws (https://arxiv.org/abs/2403.16952) which shows that loss is a somewhat linear function of mixture proportions.

Quality:
- In line 421 why use the natural data distribution for the first 500B tokens? This makes the comparison between TikMix and other static mixes a bit unfair; instead I would expect something where either 1) you use TikMix for both stage 1 and stage 2 mixture computation 2) you use an existing method to optimize the mix for Stage 1, and then show that using TikMix to dynamically adjust that mix is beneficial (for instance, Aioli https://arxiv.org/abs/2411.05735 somewhat examines both of these settings, see their Table 3).  Also, I could not find a definition of natural distribution anywhere in the paper so it's unclear to me if it is even benchmarked in Table 1 or somewhere else.
- The results in Figure 3 and the Appendix do show that influence is changing as model training progresses. However, influence decreases for many domains, which makes sense. The more interesting thing is some notion of relative changes---for instance, how often is it that domain A is more influential than domain B at 0B, but at 500B this is inverted? These inversions and changes in __rank__ are what will make the optimal mix per stage vary significantly. It would be good to highlight how the ranking of influences across domains changes over time.
- The paper would be stronger if the approach was demonstrated on another set of domains; the current results are on RefinedWeb, but there may be different behaviors with more code-heavy mixes, newer pre-training datasets, etc.

Clarity:
- How large is N in line 311?
- Would appreciate more explanation of how the group influence score is computed; e.g., what are f and $\theta^\star$ in equation 8, what values do you cache vs recompute for every proposed mix, and what validation set you use.
- What are the actual weights used?

Significance:
- Overall, the paper falls a bit short in both practical relevance and scientific insight.
    - Practically, the results in Table 1 are strong, but the decision to use the natural data distribution for the first 500B tokens raises concerns about the fairness of the comparisons and whether the reported gains can truly be attributed to TikMix.
    - Scientifically, the core question is how much online (dynamic) mixing actually improves over static mixing; that is, how different is perf(p* dynamic) vs perf(p* static), and how does this depend on the content of the domains themselves? Multi-stage mixing is increasingly common, but it remains unclear whether it offers genuine benefits in terms of the **upper bound** or simply compensates for shortcomings in current static mixture methods.

**Questions:**

1. How does TikMix conceptually and empirically differ from existing online or adaptive mixing approaches?
2. Why was the natural data distribution used for the first 500B tokens in the main experiments? How is this distribution defined, and is it included or benchmarked in Table 1?
3. How would results look if you used TikMix for both Stage 1 and Stage 2 and/or if you used an existing mixing algorithm for Stage 1 and TikMix to readjust weights for Stage 2?
4. Have you tested TikMix on domain sets beyond RefinedWeb to assess generality?
5. Can you elaborate on Figure 3/the histograms in the Appendix by also reporting rank correlation between the initial and 500B influences?
6. In line 311, what is the value of N?
7. Could you elaborate more on how the group influence score is computed (Eq. 8)? Specifically, what are f and $\theta^\star$, what values are recomputed for every candidate mix, and what validation set is used?
8. What are TikMix's learned mixture weights?

---

### Note · Authors · 2026-01-06

I have read and agree with the venue's withdrawal policy on behalf of myself and my co-authors.